# Acellular Human Placenta Small-Diameter Vessels as a Favorable Source of Super-Microsurgical Vascular Replacements: A Proof of Concept

**DOI:** 10.3390/bioengineering10030337

**Published:** 2023-03-07

**Authors:** Florian Falkner, Simon Andreas Mayer, Benjamin Thomas, Sarah Onon Zimmermann, Sonja Walter, Patrick Heimel, Wilko Thiele, Jonathan Paul Sleeman, Amir Khosrow Bigdeli, Herbert Kiss, Bruno Karl Podesser, Ulrich Kneser, Helga Bergmeister, Karl Heinrich Schneider

**Affiliations:** 1Department of Hand, Plastic and Reconstructive Surgery, BG Trauma Center Ludwigshafen, University of Heidelberg, 69117 Heidelberg, Germany; 2Center for Biomedical Research and Translational Surgery, Medical University of Vienna, 1090 Vienna, Austria; 3Ludwig Boltzmann Institute for Experimental and Clinical Traumatology, 1200 Vienna, Austria; 4Core Facility Hard Tissue and Biomaterial Research, Karl Donath Laboratory, University Clinic of Dentistry, Medical University of Vienna, 1090 Vienna, Austria; 5Department of Microvascular Biology and Pathobiology, European Center for Angioscience (ECAS), Medical Faculty Mannheim, University of Heidelberg, 68167 Mannheim, Germany; 6Institute for Biological and Chemical Systems, Karlsruhe Institute of Technology, Campus North, 76131 Karlsruhe, Germany; 7Department of Obstetrics and Gynecology, Division of Obstetrics and Feto-Maternal Medicine, Medical University of Vienna, 1090 Vienna, Austria; 8Ludwig Boltzmann Institute for Cardiovascular Research, 1090 Vienna, Austria; 9Austrian Cluster for Tissue Regeneration, 1200 Vienna, Austria

**Keywords:** human placenta, small diameter vascular grafts, arteriovenous loop, super-microsurgery

## Abstract

In this study, we aimed to evaluate the human placenta as a source of blood vessels that can be harvested for vascular graft fabrication in the submillimeter range. Our approach included graft modification to prevent thrombotic events. Submillimeter arterial grafts harvested from the human placenta were decellularized and chemically crosslinked to heparin. Graft performance was evaluated using a microsurgical arteriovenous loop (AVL) model in Lewis rats. Specimens were evaluated through hematoxylin-eosin and CD31 staining of histological sections to analyze host cell immigration and vascular remodeling. Graft patency was determined 3 weeks after implantation using a vascular patency test, histology, and micro-computed tomography. A total of 14 human placenta submillimeter vessel grafts were successfully decellularized and implanted into AVLs in rats. An appropriate inner diameter to graft length ratio of 0.81 ± 0.16 mm to 7.72 ± 3.20 mm was achieved in all animals. Grafts were left in situ for a mean of 24 ± 4 days. Decellularized human placental grafts had an overall patency rate of 71% and elicited no apparent immunological responses. Histological staining revealed host cell immigration into the graft and re-endothelialization of the vessel luminal surface. This study demonstrates that decellularized vascular grafts from the human placenta have the potential to serve as super-microsurgical vascular replacements.

## 1. Introduction

During microsurgery, vascular grafts are often needed to repair vascular defects and to restore blood flow or are required in situations where locoregional recipient vessels cannot be used for microvascular anastomoses [1,2,3,4]. The current gold standard remains the harvesting of autologous arterial or venous grafts from the patient’s own vessels, due to their biocompatibility and best patency rates. However, this method has several disadvantages. For example, it requires graft material to be taken from multiple donor sites, which causes damage to healthy tissues and is often time-consuming because of the additional harvesting processes [5]. Synthetic vascular grafts have been used as an alternative in microsurgical applications, but with limited success. In particular, small caliber prostheses with inner diameters below 6 mm achieve only mediocre patency due to their thrombogenicity and poor biomechanical properties [6,7]. Alternative graft processing methods using biocompatible tissue-engineered vascular grafts could potentially overcome these limitations. However, these applications are usually technically demanding, costly, and time-consuming [8].

Recent studies have established that natural decellularized tissues have structural and mechanical properties that support host-cell migration and tissue remodeling. Importantly, the decellularization process reduces immunogenicity to an absolute minimum, and the grafts can be stored for later use [9,10,11,12]. These beneficial properties of decellularized natural biomaterials are due to the composition of their extracellular matrix and inherent bioactivity, affording them significant advantages over synthetic materials. However, the choice of an appropriate source of tissue for the production of natural decellularized tissues remains a challenge. For example, the use of animal material can lead to chronic graft rejection due to immunogenicity problems [13] and a number of ethical issues need to be considered when producing grafts from animals, or humans after cadaver donations.

The use of the human placenta as a source of graft material resolves many of these problems. Placenta is a young tissue of human origin, and accumulates in large quantities as clinical waste after childbirth, making it arguably the most easily accessible source of human allogeneic vascular tissue. It has a consistent quality, and its harvest and use does not cause harm to the donor or raise ethical concerns [14,15]. Donor material is taken exclusively from healthy middle-aged female mothers who agree to donate their clinical waste tissue after childbirth. Any pathologies or abnormalities that arise during pregnancy are considered an exclusion criterion for the donor material. This measure is intended to prevent the use of pathologic placental tissue, especially when it may compromise the integrity of vascular grafts, such as in the case of Tenney-Parker alterations [16].

In previous studies, we reported that decellularized placental chorionic arteries with an inner diameter of 2 mm and covalently modified with heparin could be used successfully for bridging vascular defects in the infrarenal aorta in a rat model, and exhibited high patency rates with no evidence of thrombus formation [17]. The placenta grafts showed low immunogenicity in vitro and in vivo. In this present study, we aimed to build on these results and investigate the hypothesis that the vasculature of the human placental chorionic plate is suitable as a biological source of human vascular allografts with a wide range of internal diameters. To this end, we set out to efficiently fabricate submillimeter decellularized and heparin-crosslinked vascular grafts and investigate their performance in a rat arteriovenous loop (AVL) model [18]. In comparison to the methods used in our previous studies that employed larger arteries with a diameter of 2 mm, only the perfusion volume was adjusted in the work presented here to account for the smaller vessel diameters, while the formulation of the solutions for the decellularization process and the incubation times remained unchanged.

Engraftment took place with a high rate of patency and was associated with a negligible immune response and re-endothelialization of the decellularized lumen. Graft modification by covalently cross-linking the matrix with heparin further improved the in vivo performance and prevented thrombotic events after implantation. To the best of our knowledge, this is the first time that arterial blood vessels from the human placenta with an inner diameter of less than 1 mm have been successfully isolated, processed, and assessed in an in vivo model for possible microsurgical applications.

## 2. Materials and Methods

### 2.1. Ethics Approval

Human placental tissues were obtained from the Department of Obstetrics and Gynecology, Medical University of Vienna, with approval from the institutional ethics committee (EK:1602/2018) and informed consent from all donors. Donor material was taken exclusively from healthy patients. Placentas were harvested after planned caesarian section deliveries at term (pregnancy week 37 + 0 to 40 + 0). The vessels were cleared of any remaining blood by perfusion with phosphate-buffered saline (PBS) solution supplemented with heparin (50 IU/mL). The whole placenta was then frozen at −80 °C until vessel isolation. All donors were serologically tested for HIV, HBV, and HCV.

### 2.2. Vessel Isolation and Matrix Graft Fabrication

The harvest of placental graft material and the decellularization process have been described previously [14]. The protocol was adapted to the small caliber of the isolated blood vessels. Briefly, arteries with inner diameters below 1 mm were isolated from the placenta chorion. Vessels were canulated and connected to 26G venflons (Figure 1A,B). To achieve decellularization, isolated blood vessels were slowly perfused with hypertonic saline solution for 1.5 h, followed by Triton X-100 (1% *v*/*v*) and ethylene-diamine-tetraacetic acid (EDTA 0.02% *w*/*w*) in PBS for 20 h. After extensive washing of the preparations with PBS, vessels were incubated with DNAse I at 4 °C overnight. For chemical sterilization, the vessels were incubated with an aqueous solution of 0.18% (*w*/*v*) peracetic acid (PAA) containing 4.8% ethanol (*v*/*v*) for 90 min [19]. To reduce the thrombogenicity of the graft material, heparin molecules were covalently linked to the decellularized grafts as previously described [20]. Briefly, vessels were incubated in a 1 M hydroxylamine sulfate aqueous solution for 15–18 h. The vessels were subsequently rinsed with dH_2_O for 30 min, then incubated for 48 h with a solution of 1-ethyl-3-(3-dimethylaminopropyl)carbodiimide (EDC) supplemented with heparin (600 IU/mL) and adjusted with HCl to pH 1.5. After heparinization, grafts were washed twice with PBS and stored in PBS with 1% penicillin-streptomycin (Sigma, Vienna, Austria) at 4 °C until further use.

### 2.3. Animal Experimentation

All experiments were approved by the animal care committee of Rheinland-Pfalz, Germany (23177-07/G 19-7-075) according to the EU Directive for animal experiments. Specific pathogen-free male Lewis rats were maintained under optimized and standardized husbandry conditions. Animals with a bodyweight of 300–400 g and a mean age of 12–16 weeks were used for AVL surgery. Ethical rules that regulate animal experimentation were followed at all times.

### 2.4. Arteriovenous Loop (AVL) Operations

All surgeries were performed by a single experienced microsurgeon. The surgical procedures were performed under inhalation anesthesia with isoflurane using a Landmark Non-Rebreathing Veterinary RTA-0011 anesthesia machine (Vetlandmedical, Louisville, KY, USA). The AVL chamber model facilitated the investigation of human placenta vessel engraftment in a high-flow system isolated from the surrounding tissue. Briefly, each AVL was generated using a decellularized human placenta vessel graft (hPG) anastomosed end-to-end between the right saphenous artery and vein with 11-0 Ethilon sutures (Ethicon, Somerville, NJ, USA) as previously published [21]. The AVL was subsequently placed into an isolation chamber and embedded in 2 layers of Matriderm^®^, an acellular dermal matrix consisting of bovine collagen and elastin (Figure 2A–F). The chamber was then closed with a lid, affixed onto the muscle fascia, and the skin above was sutured. Chambers were custom-made using heat-resistant Teflon^®^ by the Karlsruhe Institute of Technology (KIT, Berlin, Germany). One side of the chamber wall was left open to serve as the entrance for the vessels. Enoxaparin (10 mg/kg) was applied for 2 days postoperatively to prevent thrombotic events.

The grafts were left in vivo for between 21–29 days to evaluate intermediate to long-term patency. On the day of explant removal, the vascular pedicle of the AVL was dissected, then vascular occlusion or patency was determined using a vascular patency test. To determine patency, the vascular segment was first emptied by using two micro-forceps, then the occluding forceps were removed and the refilling velocity was estimated. An adequate refill velocity for both vessels confirmed graft patency (supplemental video). After the patency test, an abdominal incision was made in the midline, the descending aorta was punctured with a 24G catheter and flushed with Ringer-heparin (100 IU/mL) solution, while the inferior caval vein was cut. For micro-computed tomography scans, a 20 mL Microfil^®^ MV-122 solution (Flow Tech Inc., Boulder, CO, USA) with 0.6 mL curing agent was applied into the descending aorta via intravascular injection. The isolation chambers were subsequently explanted en bloc, and specimens were stored at 4 °C overnight before further analysis. Euthanasia of animals at the end of the experiments was achieved by intracardial injection of pentobarbital under deep anesthesia.

### 2.5. Histology

Specimens were fixed in 4% paraformaldehyde (PFA), dehydrated and paraffin-embedded. Histological sections of 5 μm thickness were obtained using a microtome. Hematoxylin-eosin (H&E) staining was performed using a standard protocol. Stained slices were visualized using conventional microscopy and recorded using Axio Vision 4 image processing software (Carl Zeiss Microscopy, White Plains, NY, USA). Nuclear staining was performed with DAPI (Thermo Fisher Scientific, Waltham, MA, USA). CD31 staining was performed using a monoclonal rabbit anti-rat antibody at a dilution of 1:100 (Abcam, Cambridge, UK) in combination with a secondary polyclonal Alexa Fluor 488-conjugated goat anti-rabbit IgG antibody (Abcam, Cambridge, UK). For antigen retrieval, sections were boiled in Epitope Retrieval Solution (IHC World, Woodstock, GA, USA).

### 2.6. Micro Computed Tomography

Microfil-perfused specimens were scanned using a 50 cabinet micro-computed tomography (µCT) scanner (SCANCO Medical AG, Brüttisellen, Switzerland) at 70 kVp and 114 µA, using a 0.5 mm Al filter with a field of view of 15.2 mm.

### 2.7. Statistical Analysis

Data was collected in excel sheets, and statistical analyses were performed using GraphPad Prism (version 9.5.0 for Windows). Dichotomous data were analyzed using the Chi-squared test, while interval-scaled data were analyzed using the unpaired *t*-test. A *p*-value of less than 0.05 was considered to be statistically significant.

## 3. Results

### 3.1. Production of Acellular Submillimeter hPGs

Placental arteries below 1.0 mm inner diameter and between 10–20 mm in length were isolated from the placental chorionic plate, decellularized, and heparinized to produce hPGs. Compared to non-decellularized graft tissue (Figure 1C,E), histological analysis of the decellularized hPGs that were used for engraftment (Figure 1D,F) showed that no nuclei remained in the preparations, verifying that the decellularization process was successful.

### 3.2. hPGs Have a Perfect Caliber Match and High Patency Rates

A total of 14 decellularized hPGs were successfully implanted into AVLs in 14 rats. An inner diameter to graft length ratio of 0.81 ± 0.16 mm to 7.72 ± 3.20 mm was achieved in all animals. The mean operation time was 4.0 ± 1.2 h. All animals survived the operative procedure without any complications caused by the anesthesia. During the postoperative course, the animals showed no signs of a foreign body reaction or systemic inflammation. The implanted isolation chamber was well tolerated without any evidence of wound healing disorders or wound fluid collection. The hPGs were left in situ for a period of 24 ± 4 days, On the day of explant removal, all hPGs were found to be integrated into the surrounding tissue. Direct vascular occlusion tests confirmed vessel patency in 10 out of 14 hPGs, giving an overall vessel patency rate of 71%. (Figure 3C, Appendix A).

### 3.3. Vascular Remodeling Takes Place in hPGs

Histological analysis confirmed the vessel patency that was observed in the macroscopic vascular occlusion tests. H&E-stained sections of the hPGs revealed a smooth inner endothelial surface in the vessel grafts without evidence of luminal narrowing, and with no signs of dilatation or aneurysm formation. Remodeling of the vascular muscle tissue by smooth muscle cells that had infiltrated the grafts was also observed. Immunohistochemical staining for CD31 revealed that immigrated endothelial cells were present on the inner surface of the lumina. Importantly, although all grafts had been infiltrated by host cells, there was no evidence of immune cell infiltration into the hPGs (Figure 4A–D). Vascular sprouting was evident in the venous segments of the AVL, but not in the hPG segments (Figure 5).

### 3.4. µCT Analysis of hPGs

No occlusions, dilatations, or luminal narrowing of the grafts in the patent animals was observed in post-mortem µCT scans (Figure 5). The change in the luminal area [δA] in all grafts was not significantly different from that in the native rat saphenous vein segment either on the day of implantation (vein segment: 0.79 ± 0.14 µm; hPG: 0.81 ± 0.16 µm, 95% CI:0.93 to 1.01 *p* = 0.11) or on the day of explant removal (vein segment: 0.80 ± 0.14 µm; hPG 0.82 ± 0.17 µm; 95% CI: 0.95 to 1.01 *p* = 0.10).

## 4. Discussion

Decellularized allogeneic or xenogenic donor vessels are an alternative to autologous vessels, especially if adequate autologous donor vessels are not available or their harvest may cause additional morbidity [22,23,24,25]. In previous studies using an experimental rat model, we demonstrated that decellularized chorionic vessels with an inner diameter of 2 mm are able to bridge vascular defects in the infrarenal aorta and that the grafts exhibit appropriate intermediate to long-term patency rates [9,22]. In the present proof-of-concept study, we investigated whether the vasculature of the placental chorionic plate has the potential to also serve as a source of tissue for human vascular allografts in the submillimeter range. We were able to isolate submillimeter-sized vessels from the chorion of the human placenta and successfully decellularize them, as evidenced by histology and nuclear staining. Using a rat AVL model in a subcutaneous isolation chamber [26], we found that chorionic vascular grafts can serve as submillimeter bypass grafts with long-term patency rates and no aneurysms or vessel rupture.

Natural allogenic biomaterials are gaining increasing attention as functional alternatives to autologous, xenogenic, and synthetic graft materials. The human placenta is the most easily accessible source of human allogenic tissue and can be obtained from healthy donors without any major restrictions. Its fetomaternal origin provides important structural proteins and growth factors that support cell-cell interactions [10,27]. The placental vascular tree consists of the umbilical cord that inserts into the chorionic plate and branches into the chorionic vasculature [28]. The chorion, therefore, offers a rich source of material for vascular allografts. Flynn et al. (2006) suggested that decellularized placental segments can be used as scaffolds for soft tissue augmentation, but did not investigate the potential of this material for vascular matrix grafts or decellularized blood vessels [29]. Our previous study was the first to show that decellularized allogenic grafts from the human placenta have the potential to be used as vascular replacement grafts [17]. The structural proteins of the vessels are preserved after the decellularization process and maintain a robust extracellular matrix structure that is resistant to systolic arterial pressures, properties that we also observed in this present study.

The experimental technique of the AVL model is challenging and thrombosis-prone because of the two super-microsurgical anastomoses that inherently increase the risk of thrombosis [30]. A retrospective study of 612 AVL operations reported a 30% thrombosis rate in this rat model, with thrombosis being associated with injury to the endothelium during vascular suturing or due to the narrowing of the vessel diameter caused by the anastomosis [31]. The patency rate of 71% we achieved here is comparable to the reported failure rates for the AVL technique with autologous vein grafts.

In the clinical context, crosslinking methods in grafts have been used to prevent coagulation due to luminal collagen exposure, and to increase their biomechanical strength. For example, umbilical cord vessels with an inner diameter of 4 mm have been crosslinked with glutaraldehyde and used for lower limb vascular repair [32,33]. Umbilical arteries that were not heparin coated showed high rates of thrombosis, which was attributed to surface thrombogenicity and caliber mismatch [34]. For this reason, alternative vessel grafts with smaller diameters that can be matched to the caliber of the recipient’s vessels would be preferable. In contrast to the umbilical vessels, arteries of the chorion can be extracted with a range of smaller diameters of between 0.6–4 mm, and thus have the potential to be matched to the vascular system of the recipient as required [30]. By using blood vessels from human placenta chorion, we were able to overcome graft-host-vessel mismatch and achieved an appropriate diameter-to-length ratio, which is recommended for graft remodeling and patency testing [35]. This observation opens up a number of possible applications in clinical practice, as smaller diameter engineered vascular grafts would be useful in numerous vascular reconstruction situations, including small or distal coronary artery bypass grafting, microvascular reconstruction procedures, neurosurgical revascularization, hand surgery, and certain pediatric vascular reconstructions.

Host immunological tolerance to biomaterials is crucial for the clinical translation of experimental allografts [36]. Recent studies have shown that macrophages play an important role in pro-inflammatory response and tissue rejection, which can be associated with poor performance outcomes [37,38,39]. We demonstrated that Triton X-100 and a novel pulsatile perfusion set-up used for the decellularization process resulted in minimal immunogenic response in vivo [17]. Our previous findings indicated that efficient decellularization is mandatory if host cell ingrowth is to be achieved in vivo, and for successful vascular graft integration and remodeling after anastomosis. In this study, we applied the same decellularization process as previously described, with flow adaptation during perfusion to maintain a luminal pressure of between 60–80 mmHg [26]. The lack of an immune response again the hPG grafts we observed here demonstrates that decellularization also elicits immunological tolerance for submillimeter arterial grafts.

A major challenge in the production of decellularized vascular grafts from native tissue is to ensure that no chemical residues remain that could have toxic effects and inhibit the immigration of host cells. By employing extensive washing steps during graft production, our decellularized chorionic vascular grafts had excellent biocompatibility, as evidenced by the ingrowth of rat endothelial cells in vivo. Vascular remodeling through extensive re-endothelialization of the luminal decellularized surface by host endothelial cells was observed, which maintained the vessel phenotypic characteristics. Nevertheless, histological examination indicated that re-endothelialization may not have been complete at the time the grafts were removed and analyzed. This may be due, for example, to a lack of adhesion sites for the endothelial cells, and suggests that additional approaches may be needed to further improve graft re-endothelialization. For example, the chemical modification of umbilical vessel grafts with glutaraldehyde has been reported to promote luminal endothelialization and vascular remodeling [40]. Chemical coupling of VEGF to graft material has also been used in efforts to enhance graft re-endothelialization [41]. Importantly, we recently found that a luminal coating of placental ECM hydrogel on ePTFE vascular grafts more strongly promotes re-endothelialization than that achieved with fibronectin coats [42], and in future work, we aim to investigate whether we can apply this method to submillimeter grafts with similar success.

In the context of tissue engineering, the creation of an AVL by interposing a vein graft between the saphenous vein and artery can be used for axial neovascularization of bioartificial tissue. Within the AVL, the autologous interposed vein graft is exposed to an arterial flow pattern, which stimulates angio-inductive properties and promotes neovascularization [21,43]. Recent studies have shown that the vein graft is the crucial effector of early angiogenesis during the neovascularization of a fibrin matrix [35]. In the present study with hPGs, it is therefore notable that we found no evidence for neovascularization that arose from the interposed graft segment. As placental arteries were used to create the hPGs, we hypothesize that the blood vessel wall may have been too thick for the proteolysis-dependent outgrowth of activated endothelial cells, rendering vessel tube formation and sprouting impossible. A longer period of implantation using venous hPGs would be needed to clarify whether hPGs also have the potential to contribute to AVL-driven neovascularization.

Despite the promising results we report, our study has limitations. First, the conclusions of the study are based on a relatively small number of samples. Nevertheless, the sample size is comparable to that reported in previously published studies in this context, and the number of animals used needed to conform to the stipulations of the Animal Welfare Act. Second, the different physiology of rats compared to humans needs to be taken into account. Nonetheless, the rat model serves as an important first step in the investigation of the biocompatibility and function of vascular grafts in vivo, prior to translation to large animal studies. We are also aware that the arterio-venous circulation model used here has different flow characteristics compared to arterio-arterial models that might better mimic the clinical situation. Third, although it would be of great interest for tissue engineering approaches if neovascularization could be induced in the hPG-AVL model, we found no evidence that the interposing arterial hPGs were able to contribute to angiogenesis, which would be crucial for the generation of large volumes of fibrovascular tissue. This is an important finding because the application of arterial hPGs in tissue engineering may not be appropriate for tissue flap generation. Venous hPGs are likely to be more suitable for the induction of angiogenesis in the AVL model, which we will investigate in future studies. Further confirmatory studies in large animal models will also be required to support translation into future clinical applications. Fourth, vascular sprouting was evident in the venous segment of the AVL in histological sections, but not in the µCT scans. We speculate that this may have been due to the viscosity of the contrast agent, and thus further dilution of the contrast agent before perfusion into the AVL may help to resolve this issue.

In future work, we plan to follow up on the results presented here in a number of ways. Biomechanical tests before and after implantation will be employed to garner more information about the remodeling of the grafts. Flow measurements using Doppler ultrasound will also be performed. Possible graft reactions to vasoconstrictive substances after implantation and an assessment of cell migration will also be considered, although this might be difficult to investigate using this model because the vessels are firmly integrated into the matrix surrounding the AVL.

In summary, by selecting appropriately-sized small-diameter vascular grafts from the human placenta chorion, we were able to overcome graft-host-vessel mismatch in the rat AVL model and achieved an appropriate diameter-to-graft length ratio. The tested hpGs showed patency rates comparable to those obtained using autologous blood vessels.

## 5. Conclusions

Here we demonstrate the potential of using placental sub-millimeter vascular grafts for microsurgical applications. This study shows for the first time that decellularized and heparinized allogeneic vascular grafts from the human placenta with a diameter of less than 1 mm can be used successfully as sub-millimeter vascular grafts. Future studies will extend these observations, with a view to possible clinical applications in tissue engineering and regeneration.

## Figures and Tables

**Figure 1 bioengineering-10-00337-f001:**
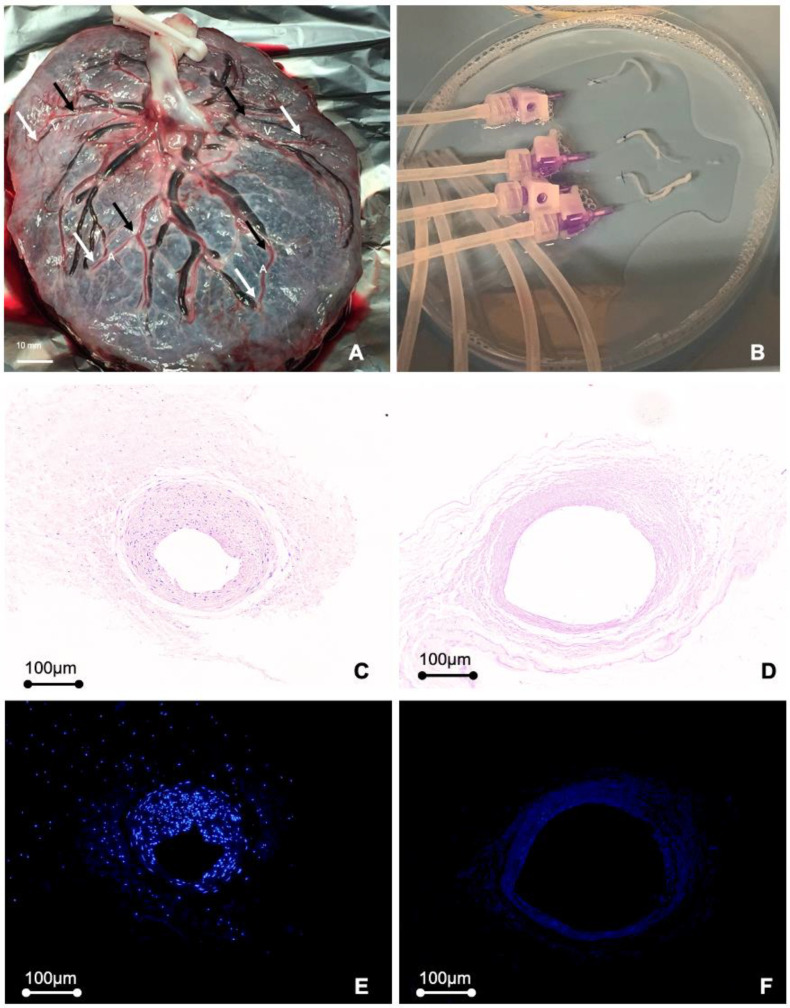
Human placental graft (hPG) isolation and the decellularization process. (**A**) Top view of the human placenta chorion, showing its distinct vasculature. (**B**) Submillimeter arteries were connected to 26G venflons for perfusion during decellularization. H&E-staining (**C**,**D**) and DAPI staining (**E**,**F**) of histological sections was used to compare native non-decellularized hPGs (**C**,**E**) with decellularized hPGs (**D**,**F**) to verify that hPGs were completely decellularized prior to implantation.

**Figure 2 bioengineering-10-00337-f002:**
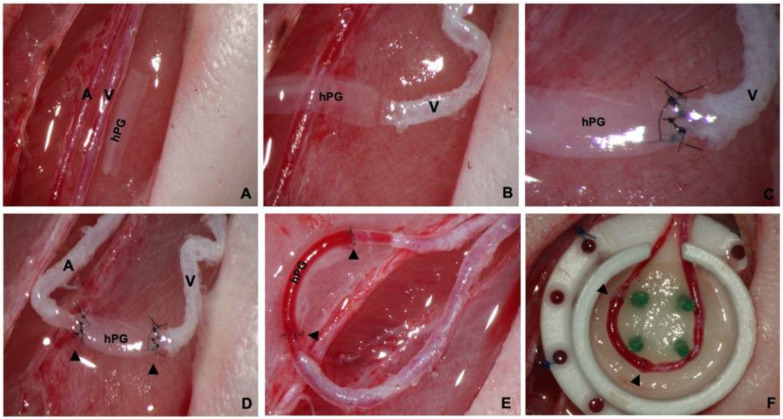
Preparation of arteriovenous loops through hPG engraftment. (**A**) After the skin incision, the saphenous artery and vein were prepared and mobilized. A human placental graft (hPG) of appropriate length and inner diameter was placed in situ. (**B**) To create an AVL an end-to-end anastomosis between the hPG and the saphenous vein was first performed. (**C**,**D**) An end-to-end anastomosis between the hPG and the saphenous artery was then performed. (**E**) After releasing the vessel clamps, the AVL filled with blood, demonstrating a perfect interpositional bypass graft of appropriate length and diameter. (**F**) The AVL was subsequently placed into an isolation chamber and embedded in a bovine collagen/elastin acellular dermal matrix (Matriderm^®^). The black triangles show the anastomosis sites.

**Figure 3 bioengineering-10-00337-f003:**
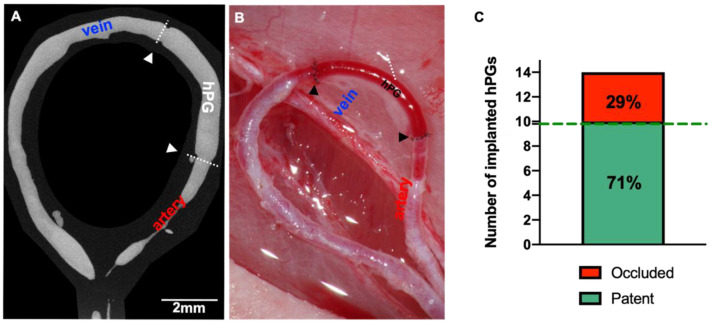
(**A**) Micro-computed tomography showing that dilation and/or luminal narrowing were not present in the grafts of patent rats. (**B**) A macroscopic view confirming the patency of the grafts. Short arrows mark the position of anastomoses. (**C**) A graphical overview of the graft patency rate was obtained during this study. The dashed line shows the average patency rate that was achieved using autologous grafts in previous studies.

**Figure 4 bioengineering-10-00337-f004:**
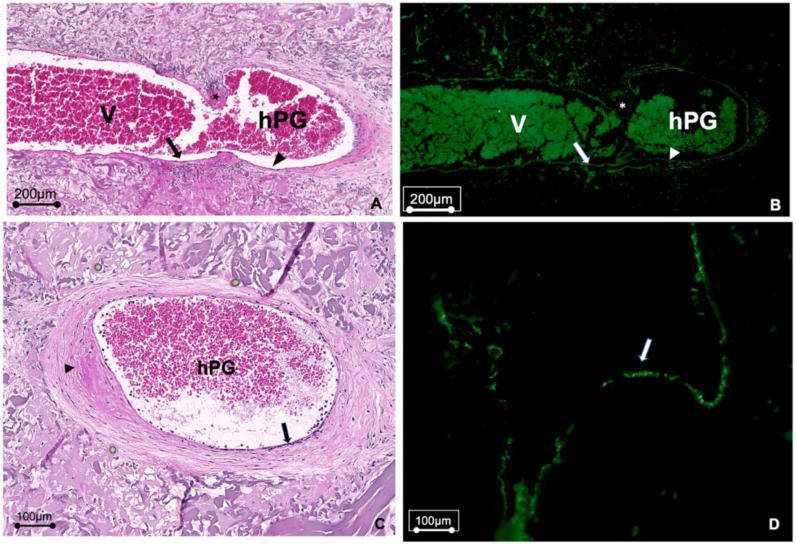
Histological examination of grafts after explant removal. (**A**,**B**) Histological sections of the end-to-end anastomosis (*****) between the saphenous vein (V) and the hPG show recellularization at the anastomosis. (**A**) H&E staining showing that the hPG was re-cellularized. (**B**) CD31 staining showing an inner lining of endothelial cells on the luminal surface that is comparable to both sides of the anastomosis, and a similar vessel wall structure in the saphenous vein and the hPG. (**C**,**D**) Histological sections through the hPG illustrating re-endothelialization of the graft lumen. (**C**) H&E staining shows a smooth endothelial surface (long arrow) without evidence of dilation or aneurysm formation. In addition, the remodeling of the vascular muscle tissue by smooth muscle cells that have infiltrated the graft (short arrow) can be observed. (**D**) CD31 staining confirming the endothelialization of the hPG (arrow).

**Figure 5 bioengineering-10-00337-f005:**
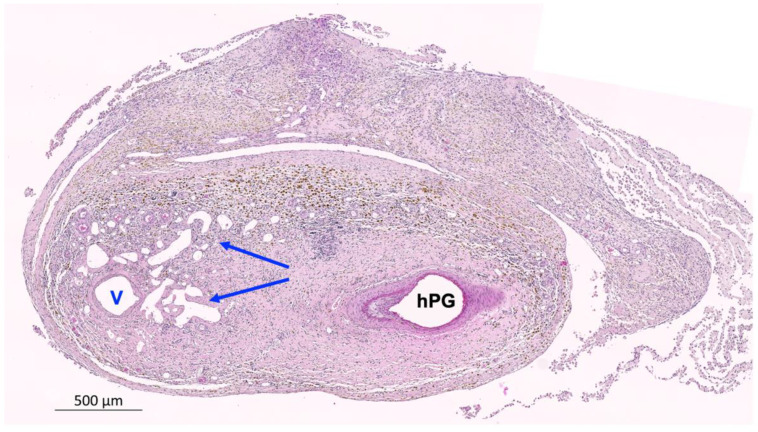
H&E staining of a histological section from an AVL explant, showing vascular sprouting (blue arrows) out of the venous segment (V) of the AVL, while angiogenesis around the arterial hPG was inapparent.

## Data Availability

Data will be made availability under request.

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
