# Peer review of "Acellular Human Placenta Small-Diameter Vessels as a Favorable Source of Super-Microsurgical Vascular Replacements: A Proof of Concept"

_bioengineering, 2023, doi:10.3390/bioengineering10030337_

Round 1

Reviewer 1 Report

The manuscript by Falkner and cols. aim to generate decellularized and heparin-linked small diameter vascular grafts obtained from placenta and to investigate their suitability in an arteriovenous loop (AVL) model in rats. In this study, arteries with inner diameters under 1 mm were isolated from the placenta chorion, decellularized and heparin molecules were covalently linked prior to re-implantation in rats. The grafts were evaluated 21 to 29 days after for patency, micro-computed tomography and excised for histological analysis. Overall patency rate of grafts was 71%, with no apparent occlusions, dilatations, or luminal narrowing. Histological analysis indicated formation of an endothelial layer and smooth muscle cell migration.

1.       Do the authors have any information regarding the donor’s blood pressure? Considering the microvascular changes in patients at risk for hypertension, metabolic disease and perhaps pre-eclampsia can the quality of the grafts obtained be impacted by these conditions?

2.       In the supplemental video can the authors provide a video demonstrating refilling velocity, or lack of, in an occluded vessel? Have the authors investigated the blood flow velocity to rule out any partially obstructed construct?

3.       Was the reason for occlusion investigated? Any sign of calcification in the constructs?

4.       Endothelization of the graft does not seem complete, would extending the study promote full repopulation of the constructs? Alternatively, can other sources of functionalization other than heparin, such as VEGF, fibronectin, RGD or other improve this process?

5.       At the end of the study are the grafts fully matured and able to respond to vasoactive agents?

6.       Can the authors provide a little more detail regarding the timelines used in the study? How long after caesarian procedure the decellularized scaffolds were implanted? How extending or reducing this timeline potentially affects the end patency of the constructs?

7.       Was there any changes in the mechanical properties of the harvested grafts when compared to pre-implantation? Any signs of collagen or other extracellular matrix deposition?

Author Response

The manuscript by Falkner and cols. aim to generate decellularized and heparin-linked small diameter vascular grafts obtained from placenta and to investigate their suitability in an arteriovenous loop (AVL) model in rats. In this study, arteries with inner diameters under 1 mm were isolated from the placenta chorion, decellularized and heparin molecules were covalently linked prior to re-implantation in rats. The grafts were evaluated 21 to 29 days after for patency, micro-computed tomography and excised for histological analysis. Overall patency rate of grafts was 71%, with no apparent occlusions, dilatations, or luminal narrowing. Histological analysis indicated formation of an endothelial layer and smooth muscle cell migration.

  1. Do the authors have any information regarding the donor’s blood pressure? Considering the microvascular changes in patients at risk for hypertension, metabolic disease and perhaps pre-eclampsia can the quality of the grafts obtained be impacted by these conditions?

The reviewer is absolutely right that hypertension in donors can change the quality of the vascular system. In blood vessel replacement the mechanical properties and the microtopography of the vascular graft are the most important factors for the patency. Grafts should mimic the anatomy of the healthy host vessels in size, shape, and compliance to provide high patency. The donor material for this study was selected exclusively from healthy patients, but blood pressure status was not monitored. However, the decellularization protocol was established in a previous study using a pulsatile perfusion system which led to vascular grafts in constant quality. In the previous study blood vessels were even treated with two different decelluarization protocols which only led to minor differences in graft patency (Schneider 2018). We believe that if the treatment with two different detergents does not affect the quality of the grafts, this will not be the case even if there are slight varying blood pressure of the donor. To better address these points, we have gone into more detail in the text about the results of our earlier studies.

  1. In the supplemental video can the authors provide a video demonstrating refilling velocity, or lack of, in an occluded vessel? Have the authors investigated the blood flow velocity to rule out any partially obstructed construct?

That´s a very good point. Unfortunately, velocity studies were not included to this study. Blood flow velocity measurements were performed earlier with the AVL model but did not prove practicable with the present model due to the extremely small vessel size. However, we are working on other methods to address this issue in the future. We have now uploaded a longer clip of the video to better show the patency of the AV loop.

  1. Was the reason for occlusion investigated? Any sign of calcification in the constructs?

The thrombosis rate is in a range expected in this animal model. (Polykandriotis, et al. 2011). Factors influencing successful outcome in the arteriovenous loop model: a retrospective study of 612 loop operations. J Reconstr Microsurg 27 (1), 11-18, doi: 10.1055/s-0030-1267385.) Most likely reasons for thrombosis include injury to the endothelium during vascular suturing or narrowing of the vessel diameter by the anastomosis. We have added this to the discussion.

  1. Endothelization of the graft does not seem complete, would extending the study promote full repopulation of the constructs? Alternatively, can other sources of functionalization other than heparin, such as VEGF, fibronectin, RGD or other improve this process?

A possible reason for incomplete re-endothelialization could be a lack of adhesion sites for the endothelial cells. Heparinization of the grafts successfully prevented thrombosis, and further local or systemic anticoagulation was not required. In previous studies, the same chemistry was used to bind not only heparin but also VEGF to enhance graft re-endothelialization (cite). However, in one of our recent studies, we used ECM vs. Fibronectin coatings on ePTFE vascular grafts to promote re-endothelialization (Rohringer&Schneider). This may be our method of choice to try in one of our next experiments. The most important question will be whether we can use this method on a submillimeter graft.

  1. At the end of the study are the grafts fully matured and able to respond to vasoactive agents?

This is also a very interesting question for us. The effect of vasoactive substances on the vessel is difficult to find out in this model, because the vessel is firmly grown into the matrix and thus not visible. Therefore, vascular function was not tested in this pilot study. However, we have planned this for the future. At the Center for Biomedical Research, Medical University of Vienna, we have a DMT myograph system that allows us to test vascular and endothelial function of blood vessels down to capillary size. After providing proof of principle with high patency rates in this study, we plan to isolate the grafts from the tissue to include this myograph test setup in our future submillimeter graft studies.

  1. Can the authors provide a little more detail regarding the timelines used in the study? How long after caesarian procedure the decellularized scaffolds were implanted? How extending or reducing this timeline potentially affects the end patency of the constructs?

Graft processing includes mechanical, chemical and enzymatic steps. Since it also includes a freeze-thaw cycle, storing grafts in a frozen state has no negative impact on the grafts, especially in the early stages of the process.

The process starts with the collection of human placental tissue from healthy donors after a cesarean section. The placental tissue is bagged and frozen until vascular preparation. For vessel preparation, the tissue is thawed and the isolated vessels are frozen againat -80 °C for at least 48 hours but up to 2 months. For decellularization, blood vessels are thawed again, connected to a pulsatile perfusion system and treated as described in the M&M section. After sterilization and heparinization, grafts can be stored in buffer solution at 4 °C for up to 4 weeks or be frozen in freezing solution (RPMI, DMSO 10%) at -80 °C for up to 4 months. The effect of heparinization has been studied in vitro for these time points in previous studies. To date, the mechanical properties have only been determined for blood vessels with 2 mm inner diameter with no significant loss in tissue strength or elasticity. However, this test could be included in future experiments for submillimeter grafts as well.

  1. Was there any changes in the mechanical properties of the harvested grafts when compared to pre-implantation? Any signs of collagen or other extracellular matrix deposition?

Some of these issues have already been addressed in one of our previous studies, in which we used 2-mm inner diameter vascular grafts in a rat model (Schneider 2018). In this study the mechanical properties of the submillimeter grafts were not investigated. Collagen or other extracellular matrix deposition was not observed. As described above, the main reason for this pilot study was to learn about the patency and main function of submillimeter grafts before including some of the important experiments mentioned by the reviewer.

Reviewer 2 Report

Found this paper to be very informative.  I have several questions.

line 204-do you believe that smooth muscle infiltrating the graft will eventually lead to wall thickening and graft closure and do you anticipate in further studies to investigate this

line 193 - did the animals show signs of  foreign body reaction or systemic inflammation or was this a typographical error

line 246- does the last 30% at the end of the sentence mean anything or is this just a typo error.

Author Response

1) line 204-do you believe that smooth muscle infiltrating the graft will eventually lead to wall thickening and graft closure and do you anticipate in further studies to investigate this

Thank you for this comment. The goal of a future study will be to follow the vessels over a longer period of time, including wall thickening and other reasons for graft failure. This study was proof of concept that these grafts remain open initially. Since this is so, we will now plan further steps.

2) line 193 - did the animals show signs of foreign body reaction or systemic inflammation or was this a typographical error

No signs of foreign body reaction or systemic inflammation was observed. We added this statement to the manuscript.

3) line 246- does the last 30% at the end of the sentence mean anything or is this just a typo error.

Thank you we have deleted this typo.

Reviewer 3 Report

This goal of the study in this manuscript was to determine if placental cells could be used as vascular grafts.  This study demonstrated that placental cells could be used for grafts, but the studies suggest that additional work needs to be done to improve graft success.  There are a few minor typographical errors listed below. Otherwise this is a well written manuscript and interesting study.

line 35:  might edit "supermicrosurgical dimensions" to "Submilimeter arterial grafts" because it is more descriptive and easier for the readers to determine what you are talking about.

line 45.  Please change "responds" to "responses" and "rendothelialization" to "re-endothelialization"

Author Response

This goal of the study in this manuscript was to determine if placental cells could be used as vascular grafts.  This study demonstrated that placental cells could be used for grafts, but the studies suggest that additional work needs to be done to improve graft success.  There are a few minor typographical errors listed below. Otherwise this is a well written manuscript and interesting study.

  • line 35:  might edit "supermicrosurgical dimensions" to "Submilimeter arterial grafts" because it is more descriptive and easier for the readers to determine what you are talking about.

Thank you, we have changed this term in the text to submillimeter graft to make it more understandable for the readers.

  • line 45.  Please change "responds" to "responses" and "rendothelialization" to "re-endothelialization"

changes have been made accordingly

Reviewer 4 Report

The authors present a very interesting manuscript, well written and with extremely relevant data. The manuscript presents an adequate state of the art and that manages to justify the innovation of this study. The methodology is clear and adequately justified. Results are presented sequentially and accurately. The discussion is coherent and allows the conclusions to be supported by the results. However, one of the points that must be justified, as it is the main point of the placenta, is the placental villus. Authors should include appropriate images of the placental villi and describe them appropriately. Authors should identify the syncytiotrophoblast, cytotrophoblast, fetal capillary, and Hofbauer macrophage. Another point that the authors must identify and justify is whether there are TENNEY-PARKER CHANGES?.

Authors should improve the use of English grammar.

A graphic summary is necessary to attract the interest of the reader.

Author Response

Authors should improve the use of English grammar.

The manuscript was completely revised by an english native speaker after embedding the revisions.

 A graphic summary is necessary to attract the interest of the reader.

A graphic summary has been created

Authors should include appropriate images of the placental villi and describe them appropriately.

The blood vessels are still isolated from the top of the chorionic plate. We hope that the revised graphic summary will clarify this. Therefore, we have not gone into more detail about the Villis in this manuscript. However, the reviewer's statement has a good point. Isolation of capillary-sized vessels from placental villi would also be interesting.

Authors should identify the syncytiotrophoblast, cytotrophoblast, fetal capillary, and Hofbauer macrophage.

These pathologies also relate to the placenta Villi. We have not gone into this because we are still isolating the blood vessels from the chorion. Furthermore, the decellularization process washes out all cells and their components from the matrix. The quality of the matrix can only be optimally tested by long-term implants. This will be one of the next steps. First we had to test the patency of these submillimeter grafts. We have now tried to address this point better in the text.

Another point that the authors must identify and justify is whether there are TENNEY-PARKER CHANGES?.

Donor material was selected exclusively from healthy patients. Graft processing includes mechanical, chemical and enzymatic steps to eliminate all cellular components from the tissue. The decellularization protocol used was established in a previous study using a pulsatile perfusion system. We can only learn from the results of the grafts we used. However, in previous studies using 2-mm identity vascular grafts, only minor differences in graft patency and in vivo outcomes were observed when comparing different decellularization protocols for graft preparation (Schneider 2018).  To our opinion by fabricating an “empty” ECM vascular scaffold, we highly reduce the risk of transmitting any of the above diseases to the new host after implantation.